# Inverse Impact of Cancer Drugs on Circular and Linear RNAs in Breast Cancer Cell Lines

**DOI:** 10.3390/ncrna9030032

**Published:** 2023-05-19

**Authors:** Anna Terrazzan, Francesca Crudele, Fabio Corrà, Pietro Ancona, Jeffrey Palatini, Nicoletta Bianchi, Stefano Volinia

**Affiliations:** 1Department of Translational Medicine, University of Ferrara, 44121 Ferrara, Italy; 2Laboratory for Advanced Therapy Technologies (LTTA), University of Ferrara, 44121 Ferrara, Italy; 3Genetics Unit, Institute for Maternal and Child Health, Scientific Institute for Research, Hospitalization and Healthcare (IRCCS) Burlo Garofolo, 34137 Trieste, Italy; 4Genomics Core Facility, Centre of New Technologies, University of Warsaw, 02-097 Warsaw, Poland; 5Centrum Nauk Biologiczno-Chemicznych (Biological and Chemical Research Centre), University of Warsaw, 02-089 Warsaw, Poland

**Keywords:** circular RNA, breast cancer, drugs

## Abstract

Altered expression of circular RNAs (circRNAs) has previously been investigated in breast cancer. However, little is known about the effects of drugs on their regulation and relationship with the cognate linear transcript (linRNA). We analyzed the dysregulation of both 12 cancer-related circRNAs and their linRNAs in two breast cancer cell lines undergoing various treatments. We selected 14 well-known anticancer agents affecting different cellular pathways and examined their impact. Upon drug exposure circRNA/linRNA expression ratios increased, as a result of the downregulation of linRNA and upregulation of circRNA within the same gene. In this study, we highlighted the relevance of identifying the drug-regulated circ/linRNAs according to their oncogenic or anticancer role. Interestingly, *VRK1* and *MAN1A2* were increased by several drugs in both cell lines. However, they display opposite effects, circ/linVRK1 favors apoptosis whereas circ/linMAN1A2 stimulates cell migration, and only XL765 did not alter the ratio of other dangerous circ/linRNAs in MCF-7. In MDA-MB-231 cells, AMG511 and GSK1070916 decreased circGFRA1, as a good response to drugs. Furthermore, some circRNAs might be associated with specific mutated pathways, such as the PI3K/AKT in MCF-7 cells with circ/linHIPK3 correlating to cancer progression and drug-resistance, or NHEJ DNA repair pathway in TP-53 mutated MDA-MB-231 cells.

## 1. Introduction

Circular RNAs (circRNAs) represent a set of RNA molecules, as defined by the structure of the single strand filament, which are generated by aberrant splicing and usually characterized by different composition in exons/introns in comparison with their cognate linear mRNA (linRNAs). CircRNAs, either non-coding or coding RNA molecules, can act as oncogenes or tumor-suppressors, and might be useful in early diagnosis or as novel targets for therapeutic approaches [1,2]. The interest in circRNAs is supported by the evidence of a relationship between their expression and the onset of pathologies, although their biological functions are still far from being defined. In cancer, and especially in breast cancer (BC), circRNAs have been associated with progression and invasion [3,4,5,6,7], through their involvement in the tumor microenvironment and in inflammation. CircRNAs displayed a role also in drug-response, after treatment with Pertuzumab, which down-modulated the expression of a circRNA coding for a novel protein in BC triple-negative phenotypes [8], or in the case of Trastuzumab in HER2^+^ cells [9].

We focused on BC using two widely adopted cell lines, MCF-7 and MDA-MB-231, representing different subtypes, in which PI3K/AKT1/mTOR axis is one of the most affected. In fact, the mutation/loss/duplication in genes within this pathway correlates with poor prognosis, and higher Akt activation, mainly in the HER2^−^ and basal-like subtypes. In contrast, low pAKT levels are generally present in luminal A tumors (and MCF-7 cells) [10]. Thus, we included inhibitors of the PI3K/Akt/mTOR axis [11], such as BYL719 and AZD5363. Moreover, we also aimed to evaluate pivotal targets, such as CDK4/6, with LEE011 and other critical cancer targets.

Therefore, we studied the effect of 14 anticancer drugs [12] on 12 circRNAs and their cognate linRNAs already described as positively or negatively related to tumorigenesis [13,14,15,16] and associable with selective pathways [14,17,18,19]. Thus, our aim was to investigate the modulation of circRNAs in BC by drugs used in therapy. Hence, we assayed a host of compounds in two of the most commonly used BC cell lines.

## 2. Results

### 2.1. Drugs Differentially Regulating the Expression of circRNAs and Cognate Linear Transcripts

Generally, dysregulated gene expression is studied in cancer upon drug stimulation without considering alternative transcript forms, such as circRNAs. For this reason, we aim to investigate the levels of circRNAs compared to their linRNAs, pointing out pharmacological agents that could interfere with this type of aberrant splicing. Focusing on circular/linear deregulation, we considered 12 circRNAs that emerge in the literature as modulated in tumor tissues or cell lines with disclosed functional roles in BC. They are listed in Table 1, which also reports indications about their expression in MCF-7 and MDA-MB-231 cells, as well as their assigned functions.

We first investigated the impact of the 14 compounds reported in Table 2. Most of these compounds are commercially available and employed in conventional chemotherapy or clinical trials for BC (or other types of tumors).

CircRNAs levels have been quantified by Reverse Transcription and quantitative polymerase chain reaction (RT-qPCR) both in MCF-7 and MDA-231 cells treated with the drugs. Four to six independent experiments were carried out in duplicate, and the expression of each circRNA and cognate linRNA has been quantified by calculating the fold change (FC) value using the 2^−ΔCT^ formula and Glyceraldehyde-3-phosphate dehydrogenase (*GAPDH*) as the reference gene. We performed a statistical analysis using both the *t*-test (two-tail paired) and the Benjamini–Hochberg (BH) correction to assess the changes in the expression of circRNAs and linRNAs by each drug. In Table 3, we have indicated the general FC determined by the impact of the respective compound and its IC_50_ value, specific for each cell line, as reported on the website https://www.cancerrxgene.org (accessed on 17 January 2023). From the data analysis in Table 3, it emerges that in MCF-7 cells AZD5363, AZD8055, and DOCETAXEL significantly altered circRNAs, with a value of adjusted (adj) *p*-values < 0.05, calculated using the BH correction, while AZD7762, GEFITINIB, DOXORUBICIN dysregulated mostly linRNAs. Finally, XL765 modified the expression of both types of transcripts. In MDA-MB-231 cells, instead, only AZD5363 changed the levels of circRNAs, while AZD7762, ERLOTINIB, GEFITINIB, GSK1070916, and DOXORUBICIN altered linRNAs’ expression. We can resume that in both cell lines AZD5363 interfered with the circRNA expression, whereas the compounds AZD7762, GEFITINIB, and DOXORUBICIN interfered preferentially with linRNA transcription. Other drugs seemed to act in a more specific manner. On the one hand, AZD8055 and DOCETAXEL drove the dysregulation of circRNAs in MCF-7 cells, including XL765, which also altered the linRNAs; on the other hand, ERLOTINIB and GSK1070916 modified linRNAs in MDA-MB-231 cells.

#### 2.1.1. circRNAs and Cognate linRNAs Modulated by Drugs in MCF-7 Cells

Focusing on a single cell line, in MCF-7 we identified eight circRNAs (circ000911, circESR1, circGFRA1, circHIPK3, circIGF1R, circMAN1A2, circNCOA3, and circVRK1) and 8 linRNAs (linIGF1R, linNCOA3, linVRK1, linBCL11B, linESR1, linSNX27, linGFRA1, and linPIK3CB) differentially expressed after exposure to drugs, quantifying the FC with respect to the untreated cells (Table 4). GSK1070916, TRAMETINIB, SCH772984, and AMG511 were ineffective in MCF-7 cells, and among the analyzed genes, *AKT2* was the only unaffected transcript. We evidenced that LEE011-modulated circNCOA3 is the only one with a statistically significant adj *p*-value.

We underlined that DOCETAXEL, ERLOTINIB, and LEE011 impacted circRNAs and not the linRNAs, decreasing their expression; conversely, AZD7762 and GEFITINIB did not affect circRNAs, but interfered with the expression of linRNAs, again reducing their levels.

Among the analyzed compounds (considering those with *p*-value < 0.05), XL765 modified all eight circRNAs, followed by AZD5363 modifying six of them, DOCETAXEL deregulating five, and AZD8055 and DOXORUBICIN changing the levels of four circRNAs. While these drugs impacted a large number of circRNAs, others altered a specific circRNA, for instance BYL719, affecting circHIPK3, ERLOTINIB with circ000911, and LEE011 dysregulating circNCOA3.

Concerning the effects of the drugs on the linRNA transcripts, XL765 and DOXORUBICIN modified five linRNAs, AZD7762 and AZD8055 modified four of them, AZD5363 three, BYL719 two, while GEFITINIB only modified linSNX27.

In conclusion, four compounds determined changes in the FC of both circRNAs and cognate linRNAs: XL765 altered both isoforms encoded by *ESR1*, *GFRA1*, *IGF1R*, and *VRK1* genes; AZD5363 those transcribed from *IGF1R* and *NCOA3*; DOXORUBICIN from *GFRA1* and *NCOA3*; finally, AZD8055 only from *VRK1*.

#### 2.1.2. circRNAs and Cognate linRNAs Modulated by Drugs in MDA-MB-231 Cells

Similar evaluations were carried out on treated MDA-MB-231 cells, quantifying the expression of circRNAs as FC, calculated with respect to the untreated control cells. Data are reported in Table 5. In this case, BYL719, DOCETAXEL, SCH772984, and TRAMETINIB did not affect the expression of the analyzed transcripts.

AMG511 and LEE011 deregulated selectively circRNAs, whereas AZD8055, ERLOTINIB, GEFITINIB, and XL765 impacted specifically on linRNAs.

We identified eight dysregulated circRNAs (circ000911, circAKT2, circAKT3, circGFRA1, circIGF1R, circMAN1A2, circNCOA3, and circVRK1), among them, three were altered by DOXORUBICIN, two by AZD5363, AZD7762, and LEE011, while circGFRA1 was modulated exclusively by AMG511 and GSK1070916. The same drugs also modulated most of the linRNAs. Ten linear transcripts (linAKT3, linBCL11B, linGFRA1, linHIPK3, linIGF1R, linMAN1A2, linNCOA3, linPIK3CB, linSNX27, and linVRK1) changed their expression after treatment, eight of them with AZD7762 and six with DOXORUBICIN, while five linRNAs changed with GSK1070916, four with AZD5363, three with ERLOTINIB and XL765; finally, GEFITINIB influenced two linRNAs and AZD8055 only linMAN1A2.

Three molecules were able to modify the expression of both the circRNAs and their cognate linRNAs: AZD5363 associated with the *AKT3* gene, AZD7762 with *NCOA3* and *VRK1*, DOXORUBICIN with *MAN1A2* and *NCOA3*.

### 2.2. Dysregulation of Specific circRNAs versus Their Cognate linRNAs in BC Cells

To clarify whether a drug differentially interfered with the expression of the circRNA and cognate linRNA, we compared the “FC circRNA/FC linRNA” ratio. We considered those significantly dysregulated (Appendix A). Thus, we pointed out circRNAs that deviated from their cognate linRNAs in a treatment-specific manner. Data analysis from MCF-7 cells is reported as boxplots in Figure 1. We selected the ratio values > 2.00. The circRNA/linRNA ratios were all favorable to circRNA and resulted from general downregulation of linRNA, with some exceptions (i.e., IGFR1) and upregulation of circRNA after the exposure to the drug. The relative excess of circRNA over linRNA was evident and statistically significant (2 × 2 contingency test, *p* = 0.001). As shown in Figure 1, AZD5363 and AZD8055 were the compounds that mostly affected the expression of the circular concerning the linear isoform, indeed three pairs were deregulated, while only one was affected by XL765. The genes involved as preferential targets of these drugs were *VRK1* and *NCOA3*, while the others, *GFRA1* and *MAN1A2* were selectively modulated by AZD5363 and AZD8055, respectively.

A similar analysis, carried out on MDA-MB-231 cells (Figure 2), showed again an upregulation of circRNA and downregulation of the linear encoded by the same gene. The relative excess of circRNA over linRNA was confirmed here as well (2 × 2 contingency test, *p* = 0.013). AZD5363 altered the expression of other genes in MDA-MB-231 (*MAN1A2* and *AKT2*) compared to MCF-7 cells (*VRK1*, *GFRA1*, and *NCOA3*).

## 3. Discussion

This study would broaden the concept of biomarkers [22,23], applying it not only to cellular contests but also to a specific pharmacological response. We selected a panel of circRNAs from genes encoding co-activators of oncogenic processes or representing noticeable regulators of the pathways involved in the pathogenesis of BC. We aimed to investigate whether they could be associated with response to treatments, using two commonly used BC cell lines, MCF-7 and MDA-MB-231, exposed to 14 compounds employed in cancer therapy. MCF-7 and MDA-MB-231 are from the luminal and basal-like subtypes, which together represent over 75% of BC tumors. The remaining BC tumors are from the HER2^+^ subtype. We focused on the HER2^−^ cell lines, in order to perform a robust statistical analysis, rather than dispersed in a shallow study.

From the evaluation of each specific RNA/drug interaction, it can be highlighted that the drugs globally impacted on circRNAs and linRNAs of the host gene, leading to a reduction of linear transcripts and a gain of the circular forms. This behavior has been reported recently following treatment with Actinomycin D, which blocks transcription and was explained as a consequence of higher stability by the circRNA [16]. Nevertheless, in MCF-7 cells we saw a relative increase of a circRNA occurring together with an increase of the cognate linRNA (IGF1R). Therefore, the higher stability of circRNA might not apply to cancer drugs or might not be the only mechanism for excess circRNA.

It is important to underline that the circRNA and the cognate linRNA may have opposite functions. Indeed, while some circular and/or linear RNAs among those we studied might inhibit processes that support tumorigenicity, others implemented drug resistance or cell motility. For this reason, we highlight the importance of the drugs that modify a single circular or linear RNA in a specific type of cell. BYL719 to circHIPK3, ERLOTINIB to circ000911 and LEE011 to circNCOA3 in MCF-7 (Table 4), while AMG511 and GSK1070916 associated with circGFRA1 in MDA-MB-231 cells (Table 5). The upregulation of circGFRA1 has been associated with poorer survival of patients, it sustains cell proliferation and displays anti-apoptotic effects in triple-negative BC through the bond of miR-34a, suggesting it is a therapeutic target [21]. Additionally, the observed downregulation of circGFRA1 by the above-mentioned drugs could represent a diagnostic biomarker of effective treatment.

Considering that increased circHIPK3 levels represent a negative event often associated with the development of drug resistance [24], so much so that it should be silenced [25].

Concerning the correlation of drug-mediated activity on a linRNA, GEFITINIB modified the expression of linSNX27, in MCF-7 (Table 4), a promoter of metastasis in BC [26,27]; while AZD8055 selectively affected linMAN1A2 in MDA-MB-231 cells (Table 5) in agreement with the observations from Fan et al. [28].

Focusing on the ratio of the circRNA/linRNA pairs, the treatment with AZD5363 affected both cell lines, whereas AZD8055 and BYL719 were selective for MCF-7 (Figure 1) and AZD7762 for MDA-MB-231 cells (Figure 2). *MAN1A2* and *VRK1* genes were dysregulated in both cell lines by different drugs. CircMAN1A1 is a marker of active cell proliferation in several types of tumors [29,30], and it is associated with cell motility, which is inhibited by its silencing [30]. In contrast, VRK1 is a kinase that phosphorylates several targets in the nucleus, and its expression is upregulated in BC and seems to promote epithelial-mesenchymal transition [31]. We detected lower levels of linVRK1 after treatments and higher levels of circVRK1. Functional studies carried out by transfection with expression vectors demonstrated the positive role of circVRK1 in several BC cells, limiting cell growth and promoting apoptosis [20]. It is reasonable to think that its increase following treatments may be associated with a good response to the drug linked to a more favorable prognosis, in addition to representing a pro-apoptotic factor. However, the contemporary upregulation of circMAN1A1, promoting cell migration, could cause cancer cells to escape.

Instead, *NCOA3*, an oncogene [32], was dysregulated by several agents only in MCF-7 cells. However, there is no evidence regarding the functions of the circular isoform, so it would be very interesting to investigate. Furthermore, several *NCOA3* polymorphisms are associated with the risk of the development of BC [33].

We point out that it is crucial to identify what are the circRNAs and cognate linRNAs modulated by anticancer agents concerning their putative functions and cell-specific mutated pathways. We summarized the pathways affected by the investigated drugs associated with the modulated circRNAs or linRNAs emerging by our data in Figure 3.

Analyzing these data in relationship with the key mutations detected in MCF-7 and MDA-MB-231 cells (listed in Appendix A in the Material and Methods section) we underline that PI3KCA mutated only in MCF-7, might amplify the effects of specific drugs for alteration of Akt pathway. In MCF-7 cells an increase of circGFRA1 [21] underlines a cell response to the treatments linked to events of cell survival and resistance [34] and might interfere with the PI3K/Akt pathway, the target of BYL719. Note that a reduction of the levels of linHIPK3, but not circHIPK3 [24,35,36,37,38], showed antagonistic regulation in cancer, in which a lowering of linHIPK3 levels correlates with lower autophagy [39], and a ratio >0.49 correlated to poor survival of patients affected by non-small cell lung cancer [39]. Anticancer drugs differently affect the thin balance of these two types of transcripts, usually leading to higher levels of circHIPK3, promoting EMT and suppressing apoptosis [40]. At the same time, they are associated with drug resistance [41]. We observed that upregulation of circHIPK3 often occurred after treatment of MCF-7 cells, in which only DOCETAXEL seems to decrease its levels without affecting the parental linear.

About the PI3K/Akt pathway, the linear isoform of the *AKT2* gene, which represents a target for therapy [42], decreased in MDA-MB-231 cells. At the same time, the corresponding circular transcript increased, but its role is far from being clarified. As occurring with circAKT2, circAKT3, whose gene encodes a protein with anti-oncogenic properties [42], was also upregulated (Table 5). AZD5363, a pan-AKT inhibitor [43], and other drugs affecting PI3K/Akt pathway modulated *VRK1* in MCF-7 cells. This gene plays a role as oncogene phosphorylating histones and several transcription factors (for example TP53, c-JUN, BANF1, and ATF2) [44,45,46,47] and regulatory proteins controlling cell proliferation and sustaining tumor growth [48], interfering with non-homologous-end joining (NHEJ) DNA repair pathway (such as KAT5) [49]. Although VRK1 blocks ChK1 and ChK2, as well as TP53 phosphorylation, circ/linVRK1 is significantly modulated by AZD7762, an inhibitor of the NHEJ DNA repair pathway, only in MDA-MB-231 cells. We underline that the deregulation of this specific pathway might depend on mutated-TP53, to which it seems attributed new unknown functions. In this case, the effects of AZD7762 could sustain the aggressiveness, occurring together with an increase also of circ/linHIPK3 ratio, correlating with poor survival in other types of tumors [39], as well as an increase of circ/linMAN1A2 associated with cell migration [30]. In MDA-MB-231 cells AURKA/B inhibitor GSK1070916 affected some RNAs in common with AZD7762. AZD7762 also regulated *BCL11B*, whose decrease might display a role as an inhibitor of cell differentiation [50], as well as *SXN27*. The latter is involved in the MAPK signaling targeted by ERLOTINIB and GEFITINIB.

Taken together, these data suggest that circRNAs and linRNAs dysregulation could be associated with gene mutations affecting selective pathways as revealed by specific inhibitors of driver genes.

Finally, we can mention that the biogenesis of circRNAs is strongly associated with epigenetic modifications, such as the H3K36me3specific histone mark [14], which differently maps in these two BC cell lines influencing, for example, the recognition by regulator proteins [51] and transcriptional factors [52]. We hypothesize that drug-mediated modifications could partially depend on this mechanism or affect it.

## 4. Materials and Methods

### 4.1. Cell Cultures and Anticancer Drugs Treatments

For this study, we have employed two human BC cell lines, the hormone-responsive luminal A MCF-7 (ER^+^/PR^+^/HER2^−^) and the triple-negative MDA-MB-231 (ER^−^/PR^−^/HER2^−^), purchased from the American Type Culture Collection (ATCC, Rockville, MD, USA). They were cultured in Dulbecco’s Modified Eagle’s Medium (DMEM, Merck, Milan, Italy) supplemented with 10% fetal bovine serum (Thermo Fisher Scientific, Monza, Italy), 2 mM L-Glutamine (Merck, Milan, Italy) and 50 U/mL penicillin plus 50 µg/mL streptomycin (Merck, Milan, Italy). The cells were grown at 37 °C in a 5% CO_2_ humidified atmosphere. MCF-7 were plated in T25 flasks at an approximate density of 6,4 × 10^5^ cells/cm^2^ and MDA-MB-231 cells at 3,6 × 10^5^ cells/cm^2^ and exposed to 14 different drugs purchased from Chemietek (Indianapolis, IN, USA) for 24 h, already described by Baldassari et al. [12] and used at the IC_50_ concentration, or 1 μM (for the compounds with IC_50_ higher than 1 μM), and AMG511 at 60 nM. The IC_50_ for most of the used compounds reported in cancerrxgene (https://www.cancerrxgene.org/, accessed on 17 January 2023) is also indicated in Table 3.

These cells were characterized by specific mutations, the most important were reported in Appendix A and extracted by COSMIC, Catalogue of Somatic Mutations in Cancer, web site: https://cancer.sanger.ac.uk/cosmic, on 17 January 2023).

### 4.2. RT-qPCR Analysis

We carried out total RNA extraction from untreated cell samples or after drug treatments using TRIzol^®^ Reagent (Merck, Milan, Italy), following the instruction provided by the supplier. Purified RNA was quantified with NanoDrop™ 1000 Spectrophotometer (Thermo Fisher Scientific, Invitrogen, Monza, Italy) and 2 µg reverse transcribed using SuperScript^®^ II Reverse Transcriptase (final concentration: RNase inhibitor 1.0 U/μL, 2.5 µM random primers, 0.5 mM of each dNTPs, 1.75 mM MgCl_2_, MultiScribe^TM^ RT 2.5 U/μL, Buffer RT 1× and 5 mM DTT) using the protocol from Thermo Fisher Scientific.

The resulting cDNA was used to perform qPCR with PowerUp SYBR Green PCR Master Mix (Thermo Fisher Scientific, Invitrogen, Monza, Italy) and specific primers, reported in Appendix A. The oligonucleotides for the amplification of the circRNAs sequences were designed using the CircPrimer 2.0 software (downloadable at the following link https://www.bio-inf.cn/ accessed on 31 January 2022) [53], and PrimerBLAST tool (https://www.ncbi.nlm.nih.gov/tools/primer-blast/ accessed on 31 January 2022), to generate the primers to amplify the linRNAs, purchased from Merck. The reactions were carried out using Thermal cycler CFX96 Touch™ Real-Time PCR Detection Systems (Bio-rad Laboratories Srl, Segrate, Milan, Italy) for 40 cycles, as follows: (i) enzyme activation: 95 °C for 3 min; (ii) denaturation: 95 °C for 20 s; (iii) annealing and extension as reported in Appendix A to amplify specific targets for 1 min. *GAPDH* was used as endogenous reference control [54] and each sample (in duplicates) was analyzed in at least three independent experiments.

Quantification of the levels of circRNAs and linRNAs was obtained using Bio-Rad CFX Manager Software and the cycle threshold (Ct) methods, determining the average of the PCR Ct duplicates. ΔCT method was used to compare each condition with the housekeeping gene (*GAPDH*) to normalize the values expressed as 2^−ΔCT^. Furthermore, we compared treated to untreated samples with the 2^−ΔΔCT^ formula.

### 4.3. Statistical Analysis

The results of PCRs were analyzed with Student’s *t*-test (two-tail paired). The *p*-values and adjusted *p*-values (calculated using the BH correction) < 0.05 were considered statistically significant.

## 5. Conclusions

The circRNAs are molecules widely studied for their still-debated cellular roles. In this study, we investigated a host of circRNAs associated with cancer. Using two cell lines, we tested various anticancer agents to define a panel of drug-modulated circRNAs/linRNAs. In general, we observed a consistent upregulation of the circRNA and a concurrent downregulation of the linRNA that may be associable with different cellular responses, drug resistance, and cell survival. Interestingly, in MCF-7 cells an exclusive increase of circ/linVRK1 mediated by XL765 indicated effective treatment without upregulation of dangerous circRNAs, such as circ/linMAN1A2, which leads to greater motility, while in MDA-MB-231 cells AMG511 and GSK1070916 1 appeared to be more specific and able to decrease circGFRA1, as a good response to drugs.

## Figures and Tables

**Figure 1 ncrna-09-00032-f001:**
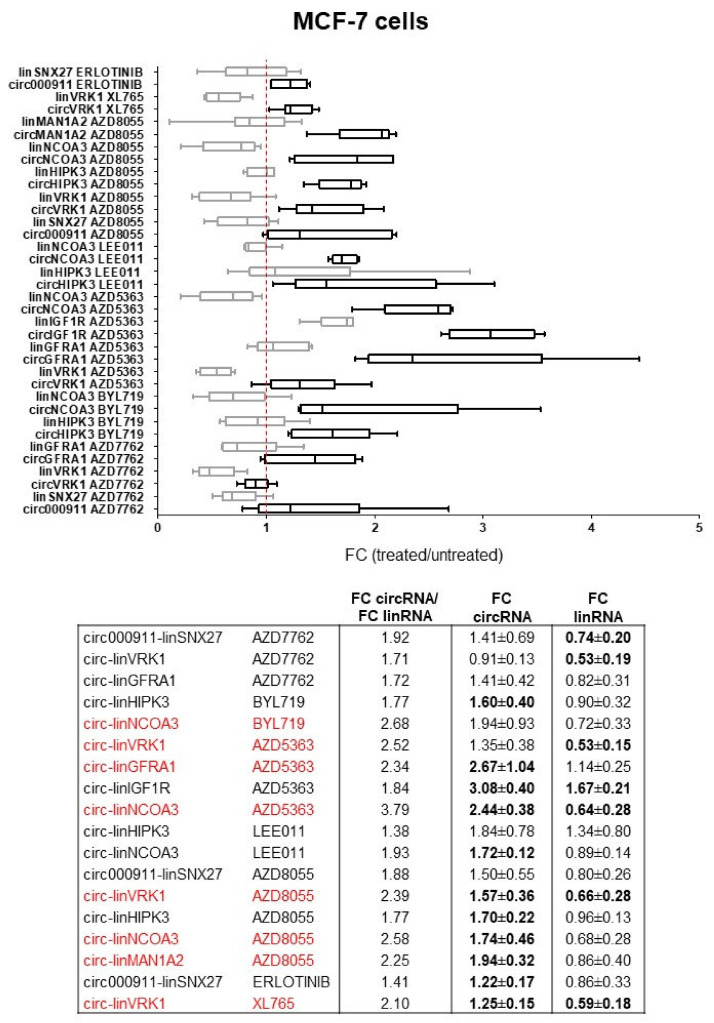
Drug-mediated dysregulation of the circRNA and linRNA pairs in MCF-7 cells. On the top, differential expression of circRNA (**black**) and linRNA (**grey**), as FC (2^−ΔΔCT^) of treated vs. untreated samples (Student’s *t*-test, two-tail paired). Boxplots were obtained using GraphPad Prism version 5.01. On the bottom, FC values and “FC circRNA/FC linRNA” ratio are reported for each pair. In bold, the values of the statistically significant FC circRNA and FC linRNA ± SD, as previously reported in Table 4. In **red**, the pairs with a ratio > 2.00.

**Figure 2 ncrna-09-00032-f002:**
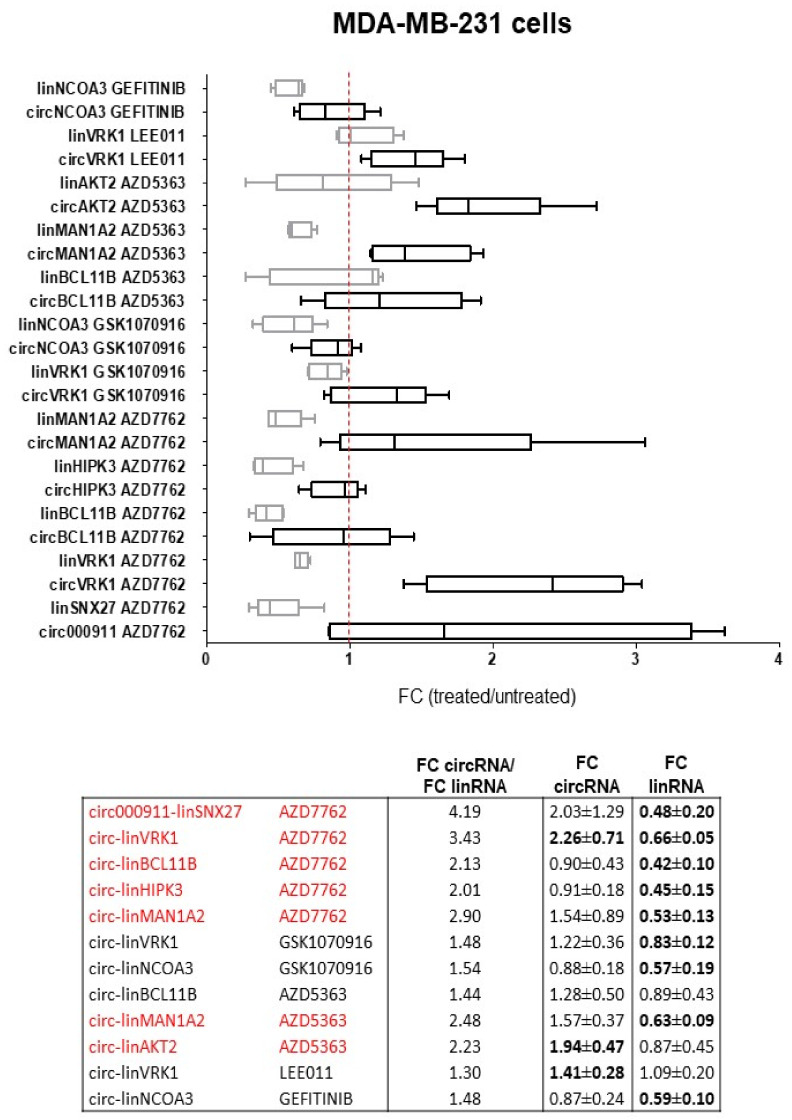
Drug-mediated dysregulation of the circRNA and linRNA pairs in MDA-MB-231 cells. On the top, differential expression of circRNA (**black**) and linRNA (**grey**), as FC (2^−ΔΔCT^) of treated vs. untreated samples (Student’s *t*-test, two-tail paired). Boxplots were obtained using GraphPad Prism version 5.01. On the bottom, FC values and “FC circRNA/FC linRNA” ratio are reported for each pair. In bold, the values of the statistically significant FC circRNA and FC linRNA ± SD, as previously reported in Table 5. In **red**, the pairs with a ratio > 2.00.

**Figure 3 ncrna-09-00032-f003:**
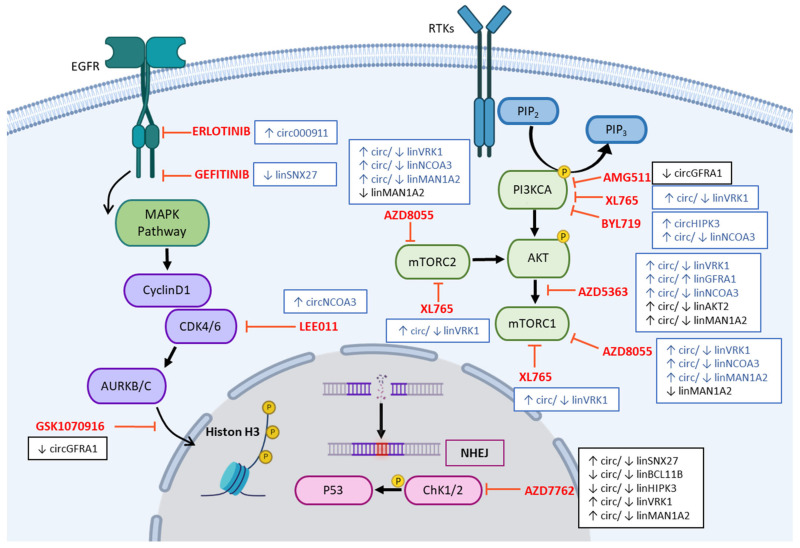
Pathways affected by drugs deregulating circRNAs and linRNAs. Some icons of this figure were created on Biorender.com (accessed on 17 January 2023). The cartoon depicts the results of circRNA and linRNA analysis linked to the pathways regulated by the indicated drug (in **red**). Pointed arrows indicate activation, while flat-tipped arrows inhibition of targets. The circ/linRNAs of each cell line are represented in the marked white squares, in **blue** the ones detected in MCF-7 and in black those in MDA-MB-231 cells.

**Table 1 ncrna-09-00032-t001:** CircRNAs expressed in BC cells and selected from the literature for our study.

circRNA	Modulation	Functions/ Processes	Reference
circIGF1R	upregulated (MCF-7 vs. MDA-MB-231 cells)	Biogenesis by H3K36me3	Coscujuela Tarrero et al., 2018 [14]
circESR1	upregulated (MCF-7 vs. MDA-MB-231 cells)	Biogenesis by H3K36me3	Coscujuela Tarrero et al., 2018 [14]
circHIPK3	similar expression (MCF-7 vs. MDA-MB-231 cells)	Biogenesis by H3K36me3	Coscujuela Tarrero et al., 2018 [14]
circNCOA3	upregulated (MCF-7 vs. MDA-MB-231 cells)	Biogenesis by H3K36me3	Coscujuela Tarrero et al., 2018 [14]
circMAN1A2	upregulated (MCF-7 vs. MDA-MB-231 cells)	Biogenesis by H3K36me3	Coscujuela Tarrero et al., 2018 [14]
circVRK1	downregulated	Inversely correlated with stemness of CSC ^1^, cell proliferation	Yan et al., 2017 [17] Li et al., 2020 [20]
circBCL11B	downregulated	Stemness of CSC ^1^	Yan et al., 2017 [17]
circ000911	downregulated	Cell proliferation, migration and invasion, acting as a sponge for miR-449a, targeting NF-κB signaling	Wang et al., 2018 [18]
circGFRA1	upregulated in triple-negative BC	Induction of cell proliferation, acting as ceRNAs ^2^ through the regulation of miR-34a, poor survival of patients	He et al., 2017 [21]
circAKT2	frequently expressed in tumors	Akt/PI3K signaling	Vo et al., 2019 [19]
circPIK3CB	frequently expressed in tumors	Akt/PI3K signaling	Vo et al., 2019 [19]
circAKT3	frequently expressed in tumors	Akt/PI3K signaling	Vo et al., 2019 [19]

^1^ CSC, cancer stem cells; ^2^ ceRNAs, competing endogenous RNAs; vs., versus.

**Table 2 ncrna-09-00032-t002:** Compounds employed in cell treatments.

Drug	Commercial Name/Company	Mechanism of Action/ Affected Pathway	Application
AZD5363	CAPIVASERTIB (AstraZeneca)	It binds all AKT isoforms inhibiting the substrates’ phosphorylation	Phase 3 study in combination with paclitaxel to treat advanced or metastatic triple negative BCs
AZD7762		ATP-competitive CHK1/2 inhibitor, affecting non-homologous end-joining (NHEJ) pathway that repairs double-strand breaks in DNA	Phase 1 study, also with gemcitabine in advanced solid tumors
AZD8055		ATP-competitive mTORC1/C2 inhibitor	Phase 1 study, patients with gliomas that have not responded to conventional treatments
BYL719	ALPELISIB (Novartis)	PI3K α-isoform (PIK3CA) specific inhibitor	Authorized by European Medicines Agency (EMA) in combination with Fulvestrant to improve survival in patients with advanced HR^+^/HER2^−^ BCs with *PIK3CA* gene mutation
ERLOTINIB	TARCEVA	It inhibits the activity of mutated and wildtype Epidermal Growth Factor Receptor (EGFR) tyrosine kinases	Approved by EMA to cure locally advanced or metastatic Non-Small Cell Lung cancer and in patients with metastatic pancreatic cancer, in combination with Gemcitabine
GEFITINIB	IRESSA	It inhibits the activity of mutated EGFR tyrosine kinases	Approved by EMA for the exclusive use in NSCL cancer with EGFR mutations
GSK1070916		ATP competitive inhibitor of Aurora B/C kinases, inhibiting the Histone H3 phosphorylation	Phase 1 study for the treatment of adult solid tumors
GSK1120212 (TRAMETINIB)	MEKINIST	Inhibitor blocking MEK 1/2 activated by mutated BRAF protein, preventing cell proliferation	Approved for medical use by EMA to treat melanoma and non-small cell lung cancer with BRAF^V600^ mutation
LEE011	RIBOCICLIB (KISQALI)	Inhibitor of Cyclin-dependent kinase 4 and 6 (CDK 4/6), activated upon binding to D cyclins	Phase 3 study in Combination with Letrozole HR^+^/HER2^−^ advanced BCs with no prior hormonal therapy
SCH772984	orally bioavailable as MK-8353 (Merck)	ATP competitive inhibitor of selective ERK1/2 suppressing MAPK pathway signaling through induction of an inactive conformation of the phosphate-binding loop and a shift of the α-C helix. It caused G1 arrest and induced apoptosis	Phase I clinical trial in patients with advanced solid tumors, in particular melanoma and colorectal cancer with BRAF^V600^ or NRAS^Q61^ somatic mutations.
DOXORUBICIN	Adriamycin^®^, as well as Rubex^®^	DNA intercalant, TOPOiso II inhibitor	It was used to treat different types of cancers, including BCs.
DOCETAXEL	TAXOTERE	It binds tubulin, promoting the assembly and blocking the disassembly of microtubules. It causes cell-cycle arrest at the G2/M phase and inhibits Vascular endothelial growth factor (VEGF)	Phase III trial of DOCETAXEL plus cisplatin in patients with stage IV NSCL cancer, and of BC, prostate, and stomach cancer
AMG511	orally bioavailable as AOBIOUS	Selective inhibitor of pan-class I phosphatidylinositol-3 kinases (PI3Kα, β, δ and γ), selective over mTOR hVPS34, DNAPK, and a broad panel of other protein kinases. It effectively inhibited pAKT	It is currently in clinical trials for cancer treatment. In animal studies it efficaciously inhibited tumor growth in PTEN-null, KRAS mutant, and HER2 amplified xenograft.
XL765	VOXATALISIB	Reversible ATP-competitive inhibitor of pan-Class I PI3K (α, β, γ, and δ) and mTORC1/mTORC2	In clinical trials for the treatment of Glioblastoma in combination therapy with Temozolomide, prostate cancer, non-Hodgkin lymphoma or chronic lymphocytic leukaemia

**Table 3 ncrna-09-00032-t003:** Compounds modifying the expression of circRNAs and linRNAs in BC cells. Drug sensitivity (IC_50_ μM) has been indicated. Adj *p*-values (calculated using the BH correction) < 0.05 were considered statistically significant. They were obtained with the Student’s *t*-test (two-tail paired) applied to all circRNAs and linRNAs, considering the 2^−ΔCT^. Whole data are included in the Appendix A (data obtained from MCF-7 cells) and Appendix A (data obtained from MDA-MB-231 cells).

	MCF-7 Cells					MDA-MB-231 Cells				
		circRNAs		linRNAs			circRNAs		linRNAs	
DRUGS	Drug Sensitivity (IC_50_ μM)	FC	Adj *p*-Value	FC	Adj *p*-Value	Drug Sensitivity (IC_50_ μM)	FC	Adj *p*-Value	FC	Adj *p*-Value
**AZD5363**	41.4	0.69	0.008	2.39	-	111.7	0.05	0.026	0.22	-
**AZD7762**	6.8	0.54	-	2.01	0.039	1.5	0.06	-	0.14	1.0 × 10^−7^
**AZD8055**	0.5	0.65	0.027	2.78	-	0.5	0.04	-	0.25	-
**BYL719**	9.5	0.67	-	2.74	-	81.9	0.04	-	0.27	-
**ERLOTINIB**	95.6	0.60	-	3.05	-	36.2	0.05	-	0.24	0.042
**GEFITINIB**	99.6	0.34	-	2.01	0.039	46.7	0.04	-	0.22	4.2 × 10^−4^
**GSK1070916**	-	0.36	-	2.34	-	-	0.03	-	0.21	2.3 × 10^−5^
**TRAMETINIB**	48.2	0.45	-	2.81	-	0.3	0.05	-	0.26	-
**LEE011**	30.1	0.66	-	3.35	-	38.4	0.05	-	0.27	-
**SCH772984**	203.2	0.38	-	2.71	-	2.6	0.05	-	0.31	-
**DOXORUBICIN**	0.1	0.71	-	3.97	0.001	0.1	0.07	-	0.46	5.0 × 10^−6^
**DOCETAXEL**	0.01	0.25	0.039	2.50	-	0.01	0.04	-	0.26	-
**AMG511**	-	0.35	-	2.79	-	-	0.04	-	0.26	-
**XL765**	91.4	0.54	0.015	3.41	0.008	75.2	0.04	-	0.26	-

**Table 4 ncrna-09-00032-t004:** Drug-mediated expression of circRNAs and linRNAs in MCF-7 cells. For each sample are indicated the values of FC, as 2^−ΔΔCT^ (quantified pairing of the treated to the untreated cells) ± standard deviation (SD), and the correspondent *p*-values, obtained with Student’s *t*-test (two-tail paired).

Drug	circRNA	FC (2^−ΔΔCT^)	*p*-Value	Adj *p*-Value	linRNA	FC (2^−ΔΔCT^)	*p*-Value	Adj *p*-Value
AZD5363	circESR1	1.93 ± 0.72	0.046	-	-	-	-	-
AZD5363	circGFRA1	2.67 ± 1.04	0.02	-	-	-	-	-
AZD5363	circHIPK3	2.02 ± 0.79	0.045	-	-	-	-	-
AZD5363	circIGF1R	3.08 ± 0.40	0.002	-	linIGF1R	1.67 ± 0.21	0.002	-
AZD5363	circMAN1A2	2.08 ± 0.58	0.01	-	-	-	-	-
AZD5363	circNCOA3	2.44 ± 0.38	0.001	-	linNCOA3	0.64 ± 0.28	0.047	-
AZD5363	-	-	-	-	linVRK1	0.53 ± 0.15	0.002	-
AZD7762	-	-	-	-	linBCL11B	0.43 ± 0.26	0.003	-
AZD7762	-	-	-	-	linESR1	0.63 ± 0.16	0.007	-
AZD7762	-	-	-	-	linSNX27	0.74 ± 0.20	0.02	-
AZD7762	-	-	-	-	linVRK1	0.53 ± 0.19	0.01	-
AZD8055	-	-	-	-	linESR1	1.93 ± 0.39	0.006	-
AZD8055	-	-	-	-	linGFRA1	1.30 ± 0.19	0.03	-
AZD8055	-	-	-	-	linIGF1R	1.59 ± 0.38	0.03	-
AZD8055	circHIPK3	1.70 ± 0.22	0.002	-	-	-	-	-
AZD8055	circMAN1A2	1.94 ± 0.32	0.003	-	-	-	-	-
AZD8055	circNCOA3	1.74 ± 0.46	0.02	-	-	-	-	-
AZD8055	circVRK1	1.57 ± 0.36	0.006	-	linVRK1	0.66 ± 0.28	0.03	-
BYL719	-	-	-	-	linESR1	1.57 ± 0.22	0.004	-
BYL719	circHIPK3	1.60 ± 0.40	0.03	-	-	-	-	-
BYL719	-	-	-	-	linVRK1	0.74 ± 0.18	0.02	-
DOCETAXEL	circESR1	0.79 ± 0.15	0.03	-	-	-	-	-
DOCETAXEL	circHIPK3	0.65 ± 0.22	0.03	-	-	-	-	-
DOCETAXEL	circIGF1R	0.72 ± 0.17	0.02	-	-	-	-	-
DOCETAXEL	circMAN1A2	0.66 ± 0.13	0.004	-	-	-	-	-
DOCETAXEL	circNCOA3	0.71 ± 0.14	0.009	-	-	-	-	-
DOXORUBICIN	-	-	-	-	linESR1	1.71 ± 0.40	0.02	-
DOXORUBICIN	circGFRA1	1.86 ± 0.67	0.046	-	linGFRA1	1.47 ± 0.30	0.02	-
DOXORUBICIN	circHIPK3	1.65 ± 0.27	0.006	-	-	-	-	-
DOXORUBICIN	-	-	-	-	linIGF1R	1.89 ± 0.49	0.02	-
DOXORUBICIN	circMAN1A2	1.73 ± 0.51	0.03	-	-	-	-	-
DOXORUBICIN	circNCOA3	1.98 ± 0.53	0.01	-	linNCOA3	1.67 ± 0.35	0.01	-
DOXORUBICIN	-	-	-	-	linSNX27	1.60 ± 0.49	0.03	-
ERLOTINIB	circ000911	1.22 ± 0.17	0.045	-	-	-	-	-
GEFITINIB	-	-	-	-	linSNX27	0.74 ± 0.25	0.0495	-
LEE011	circNCOA3	1.72 ± 0.12	0.0002	0.04	-	-	-	-
XL765	circ000911	1.61 ± 0.51	0.03	-	-	-	-	-
XL765	circESR1	1.55 ± 0.33	0.02	-	linESR1	2.07 ± 0.45	0.006	-
XL765	circGFRA1	2.01 ± 0.66	0.03	-	linGFRA1	1.51 ± 0.15	0.002	-
XL765	circHIPK3	1.65 ± 0.43	0.03	-	-	-	-	-
XL765	circIGF1R	2.04 ± 0.62	0.02	-	linIGF1R	2.18 ± 0.78	0.03	-
XL765	circMAN1A2	1.73 ± 0.32	0.007	-	-	-	-	-
XL765	circNCOA3	1.71 ± 0.37	0.01	-	-	-	-	-
XL765	-	-	-	-	linPIK3CB	1.40 ± 0.23	0.02	-
XL765	circVRK1	1.25 ± 0.15	0.005	-	linVRK1	0.59 ± 0.18	0.01	-

**Table 5 ncrna-09-00032-t005:** Drug-mediated expression of circRNAs and linRNAs in MDA-MB-231 cells. For each sample are indicated the values of FC, as 2^−ΔΔCT^ (quantified pairing of the treated to the untreated cells) ± SD, and the corresponding significant *p*-values, obtained with Student’s *t*-test (two-tail paired).

Drug	circRNA	FC (2^−ΔΔCT^)	*p*-Value	Adj *p*-Value	linRNA	FC (2^−ΔΔCT^)	*p*-Value	Adj *p*-Value
AMG511	circGFRA1	0.71 ± 0.12	0.01	-	-	-	-	-
AZD5363	circAKT2	1.94 ± 0.47	0.01	-	-	-	-	-
AZD5363	circAKT3	2.37 ± 0.95	0.03	-	linAKT3	1.33 ± 0.19	0.02	-
AZD5363	-	-	-	-	linMAN1A2	0.63 ± 0.09	0.004	-
AZD5363	-	-	-	-	linPIK3CB	0.74 ± 0.16	0.02	-
AZD5363	-	-	-	-	linVRK1	0.68 ± 0.03	0.0003	0.016
AZD7762	-	-	-	-	linBCL11B	0.42 ± 0.10	0.00003	0.006
AZD7762	-	-	-	-	linGFRA1	0.23 ± 0.13	0.0002	0.016
AZD7762	-	-	-	-	linHIPK3	0.45 ± 0.15	0.001	0.033
AZD7762	-	-	-	-	linIGF1R	0.47 ± 0.11	0.0004	0.017
AZD7762	-	-	-	-	linMAN1A2	0.53 ± 0.13	0.0014	0.036
AZD7762	circNCOA3	0.74 ± 0.19	0.04	-	linNCOA3	0.57 ± 0.19	0.01	-
AZD7762	-	-	-	-	linSNX27	0.48 ± 0.20	0.004	-
AZD7762	circVRK1	2.26 ± 0.71	0.02	-	linVRK1	0.66 ± 0.05	0.0001	0.011
AZD8055	-	-	-	-	linMAN1A2	0.86 ± 0.03	0.0007	0.025
DOXORUBICIN	-	-	-	-	linGFRA1	1.66 ± 0.52	0.046	-
DOXORUBICIN	-	-	-	-	linHIPK3	1.58 ± 0.36	0.02	-
DOXORUBICIN	circIGF1R	1.65 ± 0.47	0.04	-	-	-	-	-
DOXORUBICIN	circMAN1A2	1.59 ± 0.20	0.003	-	linMAN1A2	1.71 ± 0.53	0.04	-
DOXORUBICIN	circNCOA3	1.65 ± 0.42	0.03	-	linNCOA3	2.08 ± 0.65	0.02	-
DOXORUBICIN	-	-	-	-	linSNX27	2.01 ± 0.55	0.02	-
DOXORUBICIN	-	-	-	-	linVRK1	1.85 ± 0.17	0.0004	0.017
ERLOTINIB	-	-	-	-	linHIPK3	0.72 ± 0.17	0.02	-
ERLOTINIB	-	-	-	-	linIGF1R	0.72 ± 0.17	0.02	-
ERLOTINIB	-	-	-	-	linMAN1A2	0.79 ± 0.08	0.0046	-
GEFITINIB	-	-	-	-	linNCOA3	0.59 ± 0.10	0.0008	0.026
GEFITINIB	-	-	-	-	linVRK1	0.79 ± 0.12	0.02	-
GSK1070916	-	-	-	-	linBCL11B	0.67 ± 0.20	0.01	-
GSK1070916	circGFRA1	0.52 ± 0.22	0.01	-	-	-	-	-
GSK1070916	-	-	-	-	linHIPK3	0.65 ± 0.22	0.02	-
GSK1070916	-	-	-	-	linIGF1R	0.74 ± 0.19	0.04	-
GSK1070916	-	-	-	-	linNCOA3	0.57 ± 0.19	0.008	-
GSK1070916	-	-	-	-	linVRK1	0.83 ± 0.12	0.03	-
LEE011	circ000911	1.48 ± 0.35	0.04	-	-	-	-	-
LEE011	circVRK1	1.41 ± 0.28	0.03	-	-	-	-	-
XL765	-	-	-	-	linGFRA1	0.78 ± 0.15	0.03	-
XL765	-	-	-	-	linNCOA3	0.89 ± 0.06	0.01	-
XL765	-	-	-	-	linVRK1	1.20 ± 0.12	0.02	-

## Data Availability

All data used to support the conclusions are included in the manuscript.

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
