# Peer review of "Inverse Impact of Cancer Drugs on Circular and Linear RNAs in Breast Cancer Cell Lines"

_ncrna, 2023, doi:10.3390/ncrna9030032_

Round 1

Reviewer 1 Report

Terrazan and coworkers presented the manuscript entitled “ Inverse impact of cancer drugs on circular and linear RNAs in 2 breast cancer cell lines ”. The study was well conducted and controlled with clear conclusions, however several important concerns must be fully addressed before potential publication in ncRNA journal.

Major point

1. authors studied the relationships between gene expression of circRNAs and their parental mRNAs in response to diverse drugs treatments. Although data is clear, the study it’s very descriptive, and no functional relevance in drug response was provided. It’s recommended to add results from functional studies beyond the evaluation of changes in expression levels. What is the relevance of the changes in expression of the circRNAs and mRNAs in response to drugs ? It results in a better response or in treatments resistance ? Please add functional data to response these questions at least for a circRNA/mRNA pair using cells with decreased or augmented expression of the cognate RNAs to define their roles in drugs response.

New data on the functions of these RNAs will provide a strong evidence of its role in therapy response which will strengthen the conclusions of the study.

Author Response

Reviewer 1

Terrazan and coworkers presented the manuscript entitled “Inverse impact of cancer drugs on circular and linear RNAs in 2 breast cancer cell lines ”. The study was well conducted and controlled with clear conclusions, however several important concerns must be fully addressed before potential publication in ncRNA journal.

Major point

  1. authors studied the relationships between gene expression of circRNAs and their parental mRNAs in response to diverse drugs treatments. Although data is clear, the study it’s very descriptive, and no functional relevance in drug response was provided. It’s recommended to add results from functional studies beyond the evaluation of changes in expression levels. What is the relevance of the changes in expression of the circRNAs and mRNAs in response to drugs? It results in a better response or in treatments resistance? Please add functional data to response these questions at least for a circRNA/mRNA pair using cells with decreased or augmented expression of the cognate RNAs to define their roles in drugs response.

New data on the functions of these RNAs will provide a strong evidence of its role in therapy response which will strengthen the conclusions of the study.

Answer: The reviewer raises a relevant point. The study of functional effects is undoubtedly the ultimate goal of our investigation. This will require a number of properly designed experiments, including preparation of expression vectors for specifically blocking the circular RNAs, but not their linear counterparts. Alternatively, siRNAs could also be used, but they function mainly in the cytoplasm, whereas sometimes circular RNAs maintain a nuclear localization. We planned already these experiments, but they will take time and efforts and by themselves represent the aim of our follow up project. Luckily and in line with our results, some functional studies from other labs have been performed and clearly demonstrate the role of some circular RNAs we identified in this paper. In particular, the function of circVRK1 was studied in breast cancer by Yang Li and Hai Li in 2020 (DOI: 10.1002/jcla.22980) using expression vectors. The authors demonstrated that an increase of circVRK1 correlated with favourable prognosis of survival, possibly by inhibiting cell growth and promoting apoptosis in MDA-MB-231 cells. Similarly, others circ/linRNAs we identified in our work described by the present manuscript, such as circMAN1A2, circHIPK3 and circGFRA1 have been investigated in other cancer models. Thus for 4 circRNAs we identified here in drug response there are already independent functional studies, hinting at their roles in cancer.

Based on the literature we have therefore included the following considerations in the manuscript:

Line 24: “In this study we highlighted the relevance of identifying the drug-regulated circ/linRNAs according to their oncogenic or anticancer role. Interestingly, VRK1 and MAN1A2 were increased by several drugs in both cell lines. However, they display opposite effects, circ/linVRK1 favours apoptosis whereas circ/linMAN1A2 stimulates cell migration, and only XL765 not altered the ratio of other dangerous circ/linRNAs in MCF-7. In MDA-MB-231 cells, AMG511 and GSK1070916 de-creased circGFRA1, as a good response to drugs. Furthermore, some circRNAs might be associated with specific mutated pathways, such as the PI3K/AKT in MCF-7 cells with circ/linHIPK3 correlating to cancer progression and drug-resistance, or NHEJ DNA repair pathway in TP-53 mutated MDA-MB-231 cells.”

Line 216: “BYL719 to circHIPK3, ERLOTINIB to circ000911 and LEE011 to circNCOA3 in MCF-7 cells (Table 4), while AMG511 and GSK1070916 associated with circGFRA1 in MDA-MB-231 cells (Table 5). The upregulation of circGFRA1 has been associated with poorer survival of patients and it sustains cell-proliferation and displays anti-apoptotic effects in triple negative BC through the bond of miR-34a, suggesting it as a therapeutic target [21]. Also, the observed down-regulation of circGFRA1 by the above-mentioned drugs could represent a diagnostic biomarker of effective treatment.”

Line 232: “MAN1A2 and VRK1 genes were dysregulated in both cell lines by different drugs. CircMAN1A1 is a marker of active cell proliferation in several types of tumor, [29,30], it is associated with cell motility, inhibited by its silencing [30]. In contrast, VRK1 is a kinase that phosphorylates several targets in the nucleus, and its expression is upregulated in BC and seems to promote epithelial mesenchymal transition (EMT) [31]. We detected lower levels of linVRK1 after treatments and an increase of circVRK1. Functional studies carried out by transfection with expression vectors demonstrated the positive role of circVRK1 in several BC cells, limiting cell growth and promoting apoptosis [20]. It is reasonable to think that its increase following treatments may be associated with a good response to the drug linked to a more favourable prognosis, in addition to represent a pro-apoptotic factor. However, the contemporary upregulation of circMAN1A1, promoting cell migration, could cause cancer cells to escape.”

Line 265: “Note that a reduction of the levels of linHIPK3, but not circHIPK3 [24,35-38], showed antagonistic regulation in cancer, in which a lowering of linHIPK3 levels correlate with low-er autophagy [39], and a ratio>0.49 correlated to poor survival of patients affected by on-small cell lung cancer [39]. Anticancer drugs differently affect the thinly balance of these two types of transcripts, usually leading to higher levels of circHIPK3, promoting EMT and suppressing apoptosis [40] and at the same time they are associated with drug-resistance [41]. We observed that upregulation of circHIPK3 occurred often after treatment of MCF-7 cells, in which only Docetaxel seems decrease its levels without affecting the parental linear.”

Line 357: “Interestingly, in MCF-7 cells an exclusive increase of circ/linVRK1 mediated by XL765 indicated effective treatment without upregulation of dangerous circRNAs, such as circ/linMAN1A2, which leads to greater motility, while in MDA-MB-231 cells AMG511 and GSK1070916 1 appeared to be more specific and able to decrease circGFRA, as a good response to drugs.”

Reviewer 2 Report

In this manuscript, the study analyzed circRNAs and linRNAs expression in two major breast cancer cell lines after various anti-cancer drug treatments. Authors designed and performed well experiments in terms of drug selection and biological/technical replicates in RT-qPCR, and addressed possible effects in cancer signaling pathways showing statistically significant ratio change between two RNA types. Lots of information with potential effects on circRNA in breast cancer pathways. Authors processed many replicates of sample cell lines and showed solid results. Significance is a bit moderate with the current results but there is clear importance of circRNA upon drug treatments

I have following comments regarding concerns on this manuscript

  1. Authors chose two major breast cancer cell lines for the study. MCF-7 (luminal A, hormone positive) and MDA-MB-231 (basal-like, triple negative) cell lines have been commonly used in breast cancer study but they are both HER2 negative. HER2 is one of the most targetable in breast cancer treatment. It would be more informative if authors can include HER2 overexpressed cell lines and its target drugs to cover major subtypes of breast cancer as addressed in the introduction. In addition, one representative cell line in each subtype might not be ideal to generally make a conclusion which authors would like to claim in the study.

  1. Although the study showed statistically significant results of circRNA/linRNA changes upon drug treatments and possible related mechanisms, the solid evidence of such dysregulation lacks description with experimental demonstration. For example, authors emphasized on circVRK1 effect by ratio change and suggested possible pathway dysregulations in several pathways. However it is hard to be convinced without showing evidence that essentially affects its downstream. To validate this, it would be better to show at least the status change in p53 phosphorylation or subset protein data related in NHEJ pathway. In addition, authors just used the public reference for IC50 to treat 14 compounds in each cell lines. It might be useful information but not recommended in experimental design. Systematic approach of RNAseq or protein analysis (eg. RPPA) should be applied for overall observation of pathway changes to assure the drug effectiveness.

Author Response

Reviewer 2

Comments and Suggestions for Authors

In this manuscript, the study analyzed circRNAs and linRNAs expression in two major breast cancer cell lines after various anti-cancer drug treatments. Authors designed and performed well experiments in terms of drug selection and biological/technical replicates in RT-qPCR, and addressed possible effects in cancer signaling pathways showing statistically significant ratio change between two RNA types. Lots of information with potential effects on circRNA in breast cancer pathways. Authors processed many replicates of sample cell lines and showed solid results. Significance is a bit moderate with the current results but there is clear importance of circRNA upon drug treatments.

I have following comments regarding concerns on this manuscript.

  1. Authors chose two major breast cancer cell lines for the study. MCF-7 (luminal A, hormone positive) and MDA-MB-231 (basal-like, triple negative) cell lines have been commonly used in breast cancer study but they are both HER2 negative. HER2 is one of the most targetable in breast cancer treatment. It would be more informative if authors can include HER2 overexpressed cell lines and its target drugs to cover major subtypes of breast cancer as addressed in the introduction. In addition, one representative cell line in each subtype might not be ideal to generally make a conclusion which authors would like to claim in the study.

Answer: HER2+ tumors are a very important and clinically relevant subtype of breast cancer, although it is also a minority overall. As the reviewer noted we had to perform a large number of RT-PCR assays in order to attain robust statistical analysis. We would have included more cell lines and more subtypes in our study, but it would have entailed budgetary restrictions. Since this associations between circRNAs and drug treatment is novel and well described here, for two of the most important cell lines in breast cancer, we believe the paper is worthwhile as it is and will allow other investigators, as well as ourselves, to extend the study to all the breast cancer cell types, for example in a larger mutational context. We explained better our main aim in the introduction and assessed the findings in the discussion.

Line 59: “Thus, our aim was that of investigating the modulation of circRNAs in breast cancer by drugs used in therapy. To this end, we assayed a host of compounds in two of the most commonly used BC cell lines.”

Line 197: “We aimed to investigate whether they could be associated with response to treatments, using two commonly used BC cell lines, MCF-7 and MDA-MB-231, exposed to 14 compounds employed in cancer therapy. MCF-7 and MDA-MB-231 are from the luminal and basal like subtypes, which together represent over 75% of BC tumors. The remaining BC tumors are from the HER2+ subtype. We focused on the HER2 negative cell lines, in order to perform a robust statistical analysis, rather than dispersed in a shallow study.”

  1. Although the study showed statistically significant results of circRNA/linRNA changes upon drug treatments and possible related mechanisms, the solid evidence of such dysregulation lacks description with experimental demonstration. For example, authors emphasized on circVRK1 effect by ratio change and suggested possible pathway dysregulations in several pathways. However, it is hard to be convinced without showing evidence that essentially affects its downstream. To validate this, it would be better to show at least the status change in p53 phosphorylation or subset protein data related in NHEJ pathway. In addition, authors just used the public reference for IC50 to treat 14 compounds in each cell lines. It might be useful information but not recommended in experimental design. Systematic approach of RNAseq or protein analysis (eg. RPPA) should be applied for overall observation of pathway changes to assure the drug effectiveness.

Answer: Possibly, an RNA-sequencing analysis might represent a valid approach to investigate drug-mediated changes. Nonetheless, the evaluation of the effects produced by 14 different compounds on two cell lines would have implied an excessive cost on our project, focused on the quantification of those circRNAs, suggested in literature to be altered in breast cancer. In our experiments, we demonstrated deregulation of the ratio of specific circ/linRNAs, which were also functionally established by a number of recent studies. In particular, Yang Li and Hai Li in 2020 (DOI: 10.1002/jcla.22980) demonstrated a correlation between circVRK1 and survival, as well as cell growth inhibition and apoptosis in MDA-MB-231 cells. The NHEJ DNA repair pathway, is significantly altered only in this cell line, in which TP53 gene is mutated. Therefore, the treatment of MDA-MB-231 cells with AZD7762 (inhibitor of the NHEJ DNA repair pathway) causes changes of other circ/linRNAs, such as increased circ/linHIPK3 ratio, which was associated with poor survival in other types of cancer, as well as the circ/linMAN1A2 ratio, which is associated with cell migration. An RNA-sequencing analysis could be applied to investigate specific responses, but it is in our opinion, due to the extensive time and cost scale, best left to an upcoming and separate study.

We have included the concepts detailed above, as follows:

Line 235: “In contrast, VRK1 is a kinase that phosphorylates several targets in the nucleus, and its expression is upregulated in BC and seems to promote epithelial mesenchymal transition (EMT) [30]. We detected lower levels of linVRK1 after treatments and an increase of circVRK1. Functional studies carried out by transfection with expression vectors demonstrated the positive role of circVRK1 in several BC cells, limiting cell growth and promoting apoptosis [20]. It is reasonable to think that its increase following treatments may be associated with a good response to the drug linked to a more favourable prognosis, in addition to represent a pro-apoptotic factor. However, the contemporary upregulation of circMAN1A1, promoting cell migration, could cause cancer cells to escape.”

Line 279: “This gene plays a role as oncogene phosphorylating histones and several transcription factors (for example TP53, c-JUN, BANF1 and ATF2) [44-47] and regulatory proteins con-trolling cell proliferation and sustaining tumor growth [48], interfering with non-homologous-end joining (NHEJ) DNA repair pathway (KAT5) [49]. Although VRK1 blocks ChK1 and ChK2, as well as TP53 phosphorylation, circ/linVRK1 is significantly modulated by AZD7762, an inhibitor of NHEJ DNA repair pathway, only in MDA-MB-231 cells. We underline that the deregulation of this specific pathway might depend on mu-tated-TP53, to which it seems attributed new unknown functions. In this case the effects of AZD7762 could sustain the aggressiveness, occurring together with an increase also of circ/linHIPK3 ratio, correlating with poor survival in other type of tumours [39], as well as an increase of circ/linMAN1A2 associated with cell migration [30]. In MDA-MB-231 cells AURKA/B inhibitor GSK1070916 affected some RNAs in common with AZD7762. AZD7762 also regulated BCL11B, whose decrease might display a role as an inhibitor of cell differentiation [50], as well as SXN27. The latter is involved in the MAPK signaling targeted by ERLOTINIB and GEFITINIB.”

Round 2

Reviewer 1 Report

Authors have successfully replied all the reviewer concerns, thus I recommend to accept the manuscript for publication in its actual form.